# Engagement of community health workers to improve immunization coverage through addressing inequities and enhancing data quality and use is a feasible and effective approach: An implementation study in Uganda

**Pamela Bakkabulindi**[1,2]*, **Immaculate Ampeire**[3], **Lillian Ayebale**[2,4], **Paul Mubiri**[1], **Marta Feletto**[5], **Simon Muhumuza**[1]

1 School of Public Health, Makerere University College of Health Sciences, Kampala, Uganda, 2 Health Support Initiatives, Kampala, Uganda, 3 Ministry of Health, Kampala, Uganda, 4 School of Statistics and Planning, Department of Population Studies Makerere University, Kampala, Uganda, 5 Global Health Consultant, Geneva, Switzerland

* bakkapam@gmail.com

## Abstract

### Background

Uganda, like many other developing countries, faces the challenges of unreliable estimates for its immunization target population. Strengthening immunization data quality and its use for improving immunization program performance are critical steps toward improving coverage and equity of immunization programs. The goal of this study was to determine the effectiveness of using community health workers (CHWs) to obtain quality and reliable data that can be used for planning and evidence-based response actions.

### Methods

An implementation study in which 5 health facilities were stratified and randomized in two groups to (i) receive a package of interventions including monthly health unit immunization data audit meetings, and defaulter tracking and linkage and (ii) to serve as a control group was conducted between July and September 2020. Immunization coverage of infants in both arms was determined by a review of records three months before and after the study interventions. In addition, key informant and in-depth interviews were conducted among facility-based health workers and CHWs respectively, at the endline to explore the feasibility of the interventions.

### Results

Overall, a total of 2,048 children under one year eligible for immunization were registered in Bukabooli sub-county by CHWs as compared to the estimated district population of 1,889

**Data Availability Statement:** All relevant data are within the paper and its Supporting Information files. Raw data set and Do file have been uploaded.

**Funding:** This study was supported by the Alliance for Health Policy and Systems Research (Alliance). The Alliance is able to conduct its work thanks to the commitment and support from a variety of funders. These include Gavi, the Vaccine Alliance contributing designated funding and support for this project, along with the Alliance's long-term core contributors from national governments and international institutions. For the full list of Alliance donors, please visit: https://ahpsr.who.int/about-us/funders. The funders had no role in study design, data collection and analysis, decision to publish, or preparation of the manuscript.

**Competing interests:** The authors have declared that no competing interests exist.

children representing a moderate variance of 8.4%. The study further showed that it is feasible to use CHWs to track and link defaulters to points of immunization services as more than two-thirds (68%) of the children defaulting returned for catch-up immunization services. At the endline, immunization coverage for the Oral Polio Vaccine third dose; Rotavirus vaccine second dose; Pneumococcal Conjugate Vaccine third dose increased in both the intervention and control health facilities. There was a decrease in coverage for the Measles-Rubella vaccine decreased in the intervention health facilities and a decrease in Bacillus Calmette–Guérin vaccine coverage in the control facilities. Difference in difference analysis demonstrated that the intervention caused a significant 35.1% increase in coverage of Bacillus Calmette–Guérin vaccine (CI 9.00–61.19; $p<0.05$)). The intervention facilities had a 17.9% increase in DTP3 coverage compared to the control facilities (CI: 1.69–34.1) while for MR, OPV3, and Rota2 antigens, there was no significant effect of the intervention.

## Conclusion

The use of CHWs to obtain reliable population estimates is feasible and can be useful in areas with consistently poor immunization coverage to estimate the target population. Facilitating monthly health unit immunization data audit meetings to identify, track, and link defaulters to immunization services is effective in increasing immunization coverage and equity.

## Background

Immunization is a known cost- effective public health intervention that saves millions of lives every year [1]. Immunization continues to play a crucial role in the achievement of sustainable development goal (SDG) three, as well as other thirteen SDGs directly or indirectly [2]. Since the global launch of the Expanded Program on Immunization (EPI) in 1974, immunization coverage has exceeded 80% among infants in several countries. However, in poor and developing countries, millions of children remain under/unimmunized [3]. Sub-Saharan African countries still struggle to reach the Diphtheria-Pertussis-Tetanus third dose (DPT3) coverage of 90% as well as providing equitable access to life-saving vaccines [4]. The national immunization coverage masks poor sub-national coverage as there is insufficient detail on local populations that are not fully vaccinated [5]. Even with strengthened routine immunization programs, the marginalized and vulnerable communities are usually left out. Inequity in the uptake of routine vaccines has contributed to the accumulation of un/under-immunized children, and this has been closely linked to periodic outbreaks of vaccine-preventable diseases. Understanding who is not immunized can answer why they are not immunized [4, 5]. Indeed, one of the goals of the 2030 global immunization agenda is addressing equity gaps by identifying children who have not received vaccines at all (zero-dose) as part of promoting coverage and equity [1]. Thus, there is advocacy for country-specific pro-equity strategies in order to achieve universal health coverage [2].

One of the pro-equity strategies is a focus on disaggregation of sub-national data which is able to highlight inequitable access and poor utilization of immunization services [2]. The availability, reliability and quality of immunization data is critical to the success of any immunization program. Poor quality data leads to unreliable projections, planning, and programmatic implementation, which ultimately undermines international and national

immunization investments [6]. Immunization data remains an underutilized resource in informed and timely decision-making, especially at health facility level despite the high investments in national health information systems and advances in information technology to improve data quality [7]. A number of developing countries have more than a 10% difference between administrative and survey coverage for the years 2011–2015 [8]. Strengthening immunization data quality and using data for improving immunization program performance are critical steps towards improving coverage and equity in Africa [9].

Ensuring a system of obtaining reliable immunization data remains a challenge. Some of the cited data challenges include an unstable denominator and a transient population [10]. Previously conducted studies have revealed inaccurate administrative denominator resulting into vaccine coverage beyond 100% and disease outbreaks in areas of high coverage [11]. Uganda, like many other developing countries faces challenges of unreliable and inadequate estimates for her immunization target population [12]. The national coverage of DPT3 of 92% masks important differences in performance between districts [12]. Most of the immunizable population targets and projections are based on data captured through institutional deliveries which were as low as 63% [13] meaning, a significant number of births in the community go unregistered. Reliance on institutional data underestimates the target population for immunizable eligible children (under-one-year) especially in the hard-to-reach areas.

Over the last decade, there has been a growing interest in introduction and re-vitalization of national CHW programs in low-and middle-income countries [14]. CHWs consist of paid or volunteer healthcare workers who work in communities outside of healthcare facilities [15]. There has been expanding engagement of CHWs to meet population health needs, address health inequities and improve access to health services [14].

In Uganda, CHWs are drawn from and expected to work in their communities [16]. In 2002, Uganda began implementing a national CHW program also known as the Village health teams (VHTs). In this model, VHTs are locally elected and are given responsibilities of caring for between 25–30 households [17]. Evidence from EPI reviews has shown key factors that contribute to a stagnated EPI performance affecting coverage and equity to include; low social community mobilization due to low VHT involvement; social economic factors including religious beliefs; long distances to health facilities and; low socio-economic status of parents/care-givers. Equity assessments have revealed underserved communities to include urban poor settlements, fishing communities, refugee settlements, religious sects, remote rural, island and communities in mountainous areas [18]. VHTs are not actively involved in the follow up of children defaulting immunization as they lack resources and tools. There is no system in place to trace and refer children for immunization. The tracing of children defaulting is reliant on the health facility staff who are limited in number, time, tools, and resources. Districts tend to rely on government national immunization days to provide catch-up vaccinations to eligible children including the zero and under-immunized children [19].

There are few studies that have quantitatively assessed role of CHWs in immunization [15]. There is little evidence of involvement of CHWs in health facility-based audit meetings that review immunization data so as to promote tracing and linking of children defaulting at their immunization schedules [20]. To fill this knowledge gap, we conducted an implementation study to assess the feasibility and use of CHWs in improving use of data, coverage, and equity of immunization services in Uganda. This operation research was based on the Ugandan EPI program needs with the aim of scaling up proven best practices to other districts.

This paper presents the use of CHWs to obtain reliable and quality data that can be used for planning as well as the use of generated health facility immunization data to increase immunization coverage and equity.

## Methods

### Study setting

The study was conducted in Mayuge district, located in eastern Uganda with a population of over twenty thousand infants (Fig 1). The district had several hard-to-reach areas including 7 islands which are habitable and a huge forest reserve with people residing there. As of 2020, the district had 12 sub-counties including two councils (Malongo, Jagusi, Bukatuube, Busakira, Imanyiro, Mpungwe, Baitagombwe, Wairasa, Kityererea, Kigandalo, Buwaya, Bukabooli, Mayuge town council, and Magama town council), 74 parishes and 512 villages. There were 3 health-sub districts and a total of 52 functional health centers (HC) including 1 hospital; 2 HCIVs; 6 HCIIIs and 43 HCIIs. The district was a low performing one, with coverage ranging from 70% for Bacillus Calmette–Guérin (BCG) vaccine to 76% for Diphtheria-Pertussis-Tetanus third dose (DPT3) vaccine and Pneumococcal Conjugate Vaccine third dose (PCV3) as of December 2019 [12]. Mayuge district also had a history of recurrent measles outbreaks each year in the last 5 years prior to the study.

One sub-county (Bukabooli) with one of lowest immunization coverage was purposively selected for the study in consultation with district health authorities. Bukabooli sub-county had 5 health facilities and a total population of 45,623, with coverage for DPT3 and Measles Rubella (MR) vaccine at 77% and 87%, respectively for the January to March 2020 period.

### Study design

The study was an implementation study with a two-arm cluster randomization design of health facilities in Bukabooli sub-county, Mayuge district. Three health facilities were randomized to receive the intervention package while two received no intervention. A cluster design was adopted for the following reasons. First, the study involved evaluation of interventions implemented at health facility level, and cluster randomized designs have been found to be more appropriate for the evaluation of interventions targeted at a group of people rather than individuals [21]. Second, this design provided protection against contamination across trial groups when the trial individuals are managed within the same setting [21]. In Bukabooli sub-county, health facilities were at least 5km apart, and this could have minimized the risk of possible contamination.

Before randomization, health facilities were stratified into two groups: those with MR coverage of less than 70% and those with MR coverage of above 70% (Fig 2). The stratification

| Total Population | Women of childbearing age | Expected Pregnancies | Expected Births | Children under one year |
|---|---|---|---|---|
| 548,600 | 126,178 | 27,430 | 26,607 | 23,590 |

| Immunization Coverage | | | | | Drop-out Rate |
|---|---|---|---|---|---|
| BCG | Measles | DPT3 | PCV3 | Rota2 | DPT1-3 |
| 70% | 74% | 76% | 76% | 68% | 21% |

**Fig 1. Demographic and immunization data for Mayuge District, Uganda as of December 2019.**

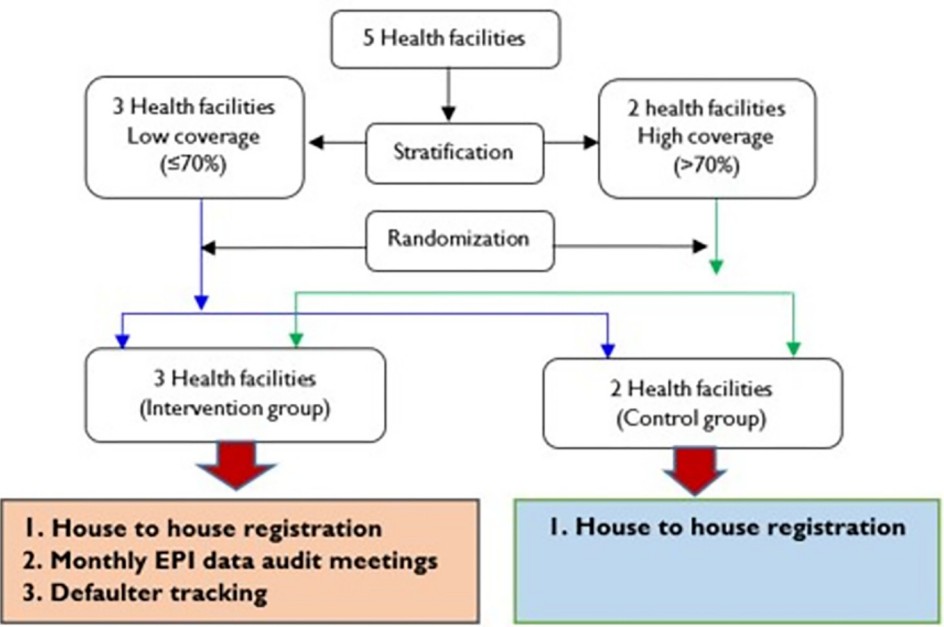

**Fig 2. Study profile showing stratification and randomization of the health facilities to the intervention and control arms in Bukabooli sub-county, Mayuge district, Uganda.**

ensured a mix of high and low coverage health facilities before randomization and a good balance of the health facility characteristics in each arm. The 5 health facility catchment areas had similar characteristics in terms of population size and distribution as well as distances to the health facilities but varied in terms of immunization coverage.

## Intervention

The package of interventions included; house-to-house registration of children under-one-year (in both intervention and control arms); health facility based audit meetings to review immunization data (in intervention arm); and defaulter tracking through home visits (in the intervention arm). The study interventions were implemented from July to September 2020. A total of 15 facility-based health workers and 51 CHWs (also locally known as village health teams who are unpaid volunteers) were oriented in the study protocol and implementation process. From each health facility, 5 health workers and 17 CHWs were chosen. In addition, they were given refresher training on basics of EPI; service delivery; data management (including review of EPI performance); vaccine management; advocacy, communication and mobilization. The training lasted for five days one month before implementation of the study interventions. The CHWs tasks included attending the data audit meetings and conducting home visits to trace and link children defaulting at their immunization schedules. They were supervised by the health workers involved in EPI service delivery at their respective health facilities to which they were attached.

The tasks of the health workers included organizing the data audit meetings; identifying children defaulting at their immunization schedules; reconciling the defaulters who had returned for immunization; and supervising the CHWs.

At baseline, for both the intervention and control arm, house-to-house registration of eligible children for immunization (0-12months) was performed as well as information on un/under-immunized children (children 0–12 and 13–23 months) who had not received/

defaulted on DPT3 and or MR was recorded by CHWs using a designed tool. The CHWs advised and encouraged caregivers of the identified un/under-immunized children to take them for catch-up vaccination at the nearest health facility. Data from this house-to-house registration was used to update the health facility target population. Immunization activities in the control arm at static and outreaches continued as per the health facility schedules.

In the intervention arm, monthly immunization data audit meetings with health facility staff and CHWs were conducted to discuss the health facility based EPI performance and generate a list of children defaulting immunization services. These immunization data audit meetings were held on monthly basis for three months at the intervention health facilities. For a given sitting, the audit meetings would last for an average of two hours. The defaulters identified through the audit meetings were then followed up and linked to health facility by CHWs through home visits. Linkage involved advising the caregiver(s) to take their defaulting children to the health facility for catch-up vaccination services. In the subsequent data audit meetings, the CHWs would report on the outcomes of tracing defaulters through home visits. In addition, they would also validate the immunization status of defaulters linked through review of the immunization registry. In the non-intervention arm, immunization activities at the static health facilities and outreaches continued as per their health facilities' schedules. CHWs were not given any remuneration. They were given airtime for communication and transport funds to facilitate their activities.

## Data collection

A structured abstraction tool was used to extract secondary data to measure primary and secondary outcome data at baseline and endline. The primary data included the total number of children under-one-year (target population) obtained through the house-to-house registration. A CHW register was developed and used to collect primary data on the target population (denominator) at baseline and was used to update the respective health facility target population in both the control and intervention arms. Health facilities adopted the house-to-house registration population data instead of the estimated projections from the district office because it was believed to be a more accurate representation. The key components of the CHW register included: location; age; vaccination status of the child; and telephone contacts of the caregiver(s).

The secondary data included the number of children that received BCG, DPT3, MR, PCV3, OPV3 and Rota2 antigens/vaccines. Data was abstracted from child immunization registers and tally sheets three months before and three months after the intervention in 5 health facilities. At both time points (baseline and endline), secondary data was abstracted from health facility records (child immunization registers, tally sheets and health facility based monthly reports and the district health management information system (DHIS). The denominator (target population) used in measurement of coverage in both arms at baseline and endline considered the data from the house-to-house registration and not the projected district population estimates.

The child immunization register is a paper-based ministry of health data collection tool found in all health facilities offering vaccination services and captures the following information; health facility name; child's name; mother's name; father's name; contact information for mother/father; address of the child; sex; date of birth; age; weight of child; dates when particular vaccines, deworming and vitamin A supplementation are administered; and status of infant screening for Human immunodeficiency virus (HIV). First time registration of children occurs as and when they attend vaccination services at the health facility.

The Ugandan vaccination schedule for children under-one-year as of the year 2020 included the following vaccines: Oral Polio vaccine (OPV); Bacille Calmette-Guérin (BCG);

Diphtheria, Pertussis, Tetanus, Hepatitis B and Haemophilus Influenza vaccine (DPT-HepB-Hib); Pneumococcal conjugate vaccine (PCV); Rotavirus vaccine; Inactivated polio vaccine (IPV); and Measles-Rubella vaccine (MR). Vaccination services are offered for free in all the Ugandan government health facilities.

## Outcome measures

The primary outcome was coverage of BCG, DPT3, MR, PCV3, OPV3 and Rota2 antigens against the target population. These antigens were selected as they are considered key to measuring the immunization performance. Target population was extracted from the primary data collected from house-to-house census of children less than 12 months.

The secondary outcome was the number defaulting children identified, followed up and returned for immunization services.

## Data management and analysis

**Quantitative data.** The quantitative data involved the data obtained from house-to-house registration; percentage of children immunized and percentage of defaulter children returning for immunization services.

The effectiveness of using CHWs to generate reliable population data (denominator) was measured by comparing the house-to-house population to the projected district population.

Using a deviation factor (%), we assessed the difference between house-to-house registration population and the district population estimates for the year 2020 which is availed by the Uganda Bureau of Statistics (UBOS) population estimates. UBOS conducted a census in 2014, however, each year national and district population data is estimated through projections. Data was analyzed at health facility level. Data analysis for variance of house-to-house population was performed in MS Excel 2013. Data verification was done to ascertain concordance between the reported and verified data. This was done using deviation factor (%) as follows:

$$\frac{(\text{house to} - \text{house} - \text{data} - \text{UBOS data}) \text{ x } 100}{\text{house} - \text{to} - \text{house data}}$$

A deviation factor of 0% implied concordance between the reported and verified data; a negative deviation factor reflects under-reporting while a positive deviation factor reflects over-reporting. Based on the guidelines which permit a variation of ± 10%, the data was categorized in three strata: (i) no variation (0≤5%) (ii) acceptable variation (5.1%-10%); (iii) excess variation (>10.1%) [11].

A dashboard showing degree of accuracy of data using color codes based on verification factors was designed as follows: **GREEN** represents acceptable variation (±5%), **YELLOW** represents moderate variation and needs improvement (-5.1% and -10% or 5.1% to 10%) and **RED** represents excessive variation and requires urgent attention (-10% or above +10%). We assessed the effectiveness of the interventions on the coverage of immunization between the intervention and control arm using the chi-squared test [11].

The immunization coverage for the different antigens based on the updated target population obtained through house-to-house registration in both the intervention and control arms was determined and compared. The coverage was measured by dividing the percentage of children vaccinated with the total updated target population.

Primary analysis was conducted with the intention to treat analyses at the health facility level. A difference in difference regression model was fitted to the data using STATA version 17. The proportion of children that received the antigen under study was computed against

the facility target population by month.

$$Y_{ij} = \beta_0 + \beta_1 period_t + \beta_2 treatment_j + \beta_3 period_t * treatment_j$$

Where $y_{ij}$ is coverage of antigen under study for health facility i in the treatment j at period t. Period is an indicator variable {1-After intervention, 0-Before intervention}, treatment is an indicator variable {1-Received the intervention, 0-Not received the intervention.

The percentage of defaulter children returning after immunization was determined by calculating the number of defaulter children returning for immunization divided by the total of number of defaulter children identified during the 3 months of the study intervention.

**Qualitative data.** To assess the feasibility (ability to carry out defined tasks) of use of CHWs; 4 Key informant interviews (KIIs) using a structured guide were conducted with one district health team (DHT) member who oversaw the CHWs and 3 health facility in-charges; and 6 In-depth Interviews (IDIs) conducted with CHWs involved in the interventions.

The KIIs and IDIs were recorded and transcribed verbatim. The textual data were complemented with additional observational notes. To familiarize oneself with the data and immerse in the details, the transcripts were read in their entirety several times. A thematic framework was identified by writing memos in form of short phrases, ideas or concepts arising from the data in the margins of the text [22]. These were then organized according to specific categories. This was followed by highlighting and sorting out quotes and making comparisons within and between cases. The quotes were then lifted from their original context and re-arranged under the newly developed themes [22]. Finally, the data was interpreted based on internal consistency, frequency and extensiveness of responses, specificity of responses and trends or concepts that cut across the various discussions. The analyzed data was presented in text form.

## Ethical considerations

Ethical clearance for the study was obtained from the Makerere University School of Public Health-Higher Degrees Research and Ethics Committee (MakSPH-HDREC). In addition, permission to conduct the research was sought from the Ministry of Health, the Mayuge District Health Office and management of the selected health facilities. Written consent was obtained from each key informant and in-depth interview respondent.

## Results

### Feasibility of using CHWs to generate reliable population data

Table 1 shows the data collected by the CHWs through house-to-house registration and compares it to the population as projected by the district using UBOS estimates [23]. Overall, 2,048 children were registered by the CHWs through house-to-house registration as compared to 1,889 according to district estimates. This represents an overall 8.4% variation that is described as moderate. In the intervention facilities, the variance between the house-to-house population and estimated district population was at 8.6% which is comparable to the 8.1% variance seen in the control facilities.

Based on the analysis of the qualitative data, we assessed and analyzed the experiences, practicability, challenges, and suggestions for improvement for house-to-house registration of eligible infants for immunization.

Overall, the health workers reported that the house-to-house registration was a practical and doable exercise which was also accepted by the community. All the in-charges of health facilities who were involved in the study appreciated the engagement of CHWs to collect

**Table 1. House-to-house registration versus the projected district population for children under-one-year in Bukabooli sub-county, Mayuge district, Uganda.**

| Health facility | House-to-House registration for under-one-year-old children | Projected district target population for under one-year-old children | Variation (%) | Color code for variation |
|---|---|---|---|---|
| **Intervention arm** | | | | |
| Buyugu HC II | 534 | 515 | 3.7% | 🟩 |
| Busira HC II | 336 | 320 | 5.0% | 🟩 |
| Nawampongo HC II | 574 | 495 | 16.0% | 🟥 |
| **Total** | **1,444** | **1,330** | **8.6%** | 🟨 |
| **Control arm** | | | | |
| Mairinya HC II | 197 | 181 | 8.8% | 🟨 |
| Bugulu HC II | 407 | 378 | 7.7% | 🟨 |
| **Total** | 599 | 559 | **8.1%** | 🟨 |
| **Overall total** | **2,048** | **1,889** | **8.4%** | 🟨 |

Note: Green represents an acceptable variation of ≤5%; Yellow represents moderate variation between -5.1% and—10% or 5.1% to 10%; Red represents an excessive variation of more than -10.1% or 10.1%.

immunization data at household level and acknowledged this was a feasible intervention as illustrated below.

The health workers (HWs) affirmed that it was one of the reliable ways of knowing the accurate target population for immunization. They further attested that it was a good experience of knowing the true location of eligible children for further follow up.

The challenges cited by the HWs were that some parents thought the registration of children were for political ambitions and/or financial gains. Indeed, some households were expecting financial reimbursement for the registration. The registration was conducted during the planting season where a few families had migrated to the forest reserves for agriculture. Other households thought that their children were going to be recruited into religious cult groups. Five known vaccine resistant households refused registration of their children.

The HWs recommended that the house-to-house registration needed to be conducted on a regular basis, at least twice a year. They emphasized the need to plan and facilitate CHWs with logistics to be able to carry out the exercise. They suggested that the community needed to be sensitized on the importance of the house-to-house registration.

*"Through the registration, I discovered that some children had even died without us knowing at the health facility", (KII with in-charge of third health facility).*

The CHWs reported that the house-to-house registration was a feasible exercise and highly acceptable by the community.

The CHWs attested that it was an important exercise because it provided an opportunity to identify children who had; missed/defaulted on their immunization schedules; died; relocated; visitors; migrants; and critically ill.

*"We discovered children who had only received one antigen like BCG and others had only received up to DPT3. Also, a number of mothers after receiving DPT3, they are told to come back at the health facility when the child is 9 months, but they never turned up. However, house to house registration helped a lot to identify children who had missed some doses like measles. For example, a child was 2 years old but looking at the health card, he had received vaccines up to DPT 3", (IDI-3 with 38-year-old CHW)*

House-to-house registration presented the opportunity for the CHWs to know their target population for their respective villages including specific location of households for these infants. CHWs were able to identify the vaccine hesitant households. They were able to know the visitors and migrants and locate families that had migrated to other locations.

They were able to sensitize the mothers on the importance of immunization and remind some /caregivers on the subsequent vaccination appointments for their children. *"Some children were visitors from another sub-county. About seven households were locked and the neighbors told us that they had gone to the forest reserve to plant maize and would return after harvesting," (IDI-5 with 42-year-old CHW)*

The challenges reported by the CHWs were that some villages were vast which required walking long distances; there were five vaccine resistant households that refused to give information on their children; it was a rainy season and yet they lacked protective gear such as raincoats, gumboots and umbrellas; there was public misconception that names of their children were written for financial/political gains; some households were hostile towards the CHWs; a few households demanded monetary payments for the registration exercise.

The CHWs recommended that they should be provided with protective gear such as umbrellas, gumboots, and raincoats to be used during harsh weather. They requested bicycles to ease their movements and t-shirts for easy identification by the community members. They requested sufficient logistics to ease their activities while in the field.

### Feasibility of using CHWs to track and link eligible children to points of service

Based on the analysis of the qualitative data, we assessed and analyzed the experiences of data harmonization, practicability, and suggestions for improvement for monthly EPI data audit meetings and home visits to determine CHWs ability to track and link eligible children to points of immunization services.

### Monthly EPI data audit meetings

Overall, there were a total of 9 data audit meetings held from July to September 2020 in the three intervention health facilities. Each facility held 3 monthly audit meetings where children defaulting at immunization services were identified, listed, and CHWs were then tasked to follow them up through home visits in the community. The health workers reported that monthly data audit meetings were important and useful as the monthly health facility-based EPI performance would be discussed thoroughly. They reported that the process for the meetings were that the health facility in-charges or EPI focal persons would identify defaulters from the child register, list them in the defaulter's register and provide lists to the CHWs for tracking and subsequent linking to the health facility. During the successive meetings, the defaulter's register would be updated and children returning for immunization identified.

*"I found the monthly meetings very useful. The defaulter tracking register was very important for tracking. Every month, after the meetings I updated the book and noted the children who had returned for immunization after being followed by the community health workers through home visits", (KII with in-charge of the first health facility).*

*"These meetings are important. We need to include these monthly meetings in our annual work-plan so that they can be funded', (KII with in-charge of the second health facility).*

The health workers reported that regular funding was needed to sustain these meetings.

Overall, CHWs reported that the monthly EPI data meetings were beneficial and feasible. They reported that the data audit meetings were very helpful in identifying children who were defaulting and ensuring each CHW obtained the particulars of these children for further follow up. The meetings were useful in updating the health workers on the progress of the work of the CHWs regarding tracking defaulters via home visits. In these meetings defaulting children who had returned for immunization were also identified and defaulter tracking register was updated.

*"Monthly data audit meeting helped to identify the children defaulting for my village catchment area and the list given to me by the in-charge is what I would use to track these children", (IDI-1 with 33-year-old CHW).*

CHWs also reported that they learnt how to approach the community to do home visits and how to conduct health education on immunization during the home visits. CHWs reported that during the study intervention they got more involved with immunization activities and appreciated their role as a bridge between the community and the health facility. In addition, they reported that the study tools used for audit meetings and defaulter tracking were helpful.

*"From these meetings I learnt a lot of things like how to communicate to the mothers/caregivers; how to deal with hostile families. I learnt a lot about immunization and the schedules, and I was able to teach the mothers during the home visits", (IDI-6 with 29-year-old CHW).*

## Home visits-defaulter tracking

After attending the monthly EPI data audit meetings, the CHWs were given a list of children defaulting in their respective villages and tasked to conduct home visits and subsequent linkage to points of service.

Overall, during the three months of the intervention, 531 children defaulting at immunization were identified during the monthly facility-based data audit meetings of which 362 (68%) returned for immunization services (Table 2). The highest return of 79% (74 of 95) was observed in Buyugu health centre II (Table 2).

Overall, the CHWs reported that the home visits were feasible and a doable exercise. They reported that a good number of mothers were not aware of the benefits of immunization.

There were some hostile households and vaccine hesitant communities where local leaders and police had to be involved.

Some of the reported challenges faced by the CHWs were that the exercise was conducted during a rainy season and yet they did not have protective gear such as raincoats. The other challenge was the hostile families that were suspicious and harsh towards the CHWs. They also complained of the long distances they needed to travel to get to some distant homes. In addition, they found that some families had migrated to other locations for farming.

**Table 2. Number of defaulter children identified, followed up, and returned for immunization services in the intervention health facilities.**

| Intervention Health Centers | Children identified and followed up (N) | Returned for immunization (N) | Returned for immunization (%) |
|---|---|---|---|
| Buyugu HC II | 95 | 74 | 79% |
| Busira HC II | 186 | 106 | 57% |
| Nawampongo HC II | 250 | 182 | 73% |
| **Total** | **531** | **362** | **68%** |

*"One of the major reasons for defaulting is migration. When the planting season starts, a number of families due to lack of land migrate to the forest reserves to be able to farm. They go for like four months and yet there are no health facilities in the forest reserves. These children will therefore default," (IDI-2 with 39-year-old CHW).*

They reported that some known vaccine hesitant households would intentionally hide their children away and lie about their vaccination status.

*"We have a problem of religious cults like triple 666, njiri-kalu and tabliqs that do not allow immunization. These cults tell their followers that immunization is bad and against their beliefs. You find that all children from these households are not immunized and even when you go to sensitize them, they hide away." (IDI-4 with 31-year-old CHW)*

The CHWs reported several reasons for children missing/defaulting from their immunization schedules. They reported that Some mothers complained about stock-out of vaccines discouraging them from returning their children for immunization. Mothers complained that children got a lot of injections which would cause swelling of the body parts and hence were hesitant to take them for immunization. Some mothers complained about being victims of domestic violence making them unable to take children for immunization. A few families migrated for various reasons such as farming for example in forest reserve areas where they were unable to access health services including immunization services. Some caregivers reported forgetting health cards at their homes when they visited new areas hence missing out on taking their children for vaccination services. Some mothers reported being busy in agricultural activities that they would forget to take their children for vaccination services. Some of the community members were ignorant about the value of immunization. The lock down due to COVID-19 pandemic with restricted movements resulted in some of children defaulting at their vaccination schedules. Some households believed in religions that forbid vaccination of human beings.

*This COVID has also brought a lot of problems. During the lock down, mothers were fearing to go to the health centers resulting in high numbers of children defaulting," (IDI-4 with 31-year-old CHW)*

The CHWs reported that home visits advanced equity as they were able to reach the hard-to-reach households including vulnerable population.

*"Through the home visits, I was able to move long distances into hard-to-reach areas such as forest reserves and islands. In one household I visited, the child was disabled and had missed most of the vaccine antigens. The mother was also lame and HIV positive. She complained that she had difficulties accessing health care due to her disabilities," (IDI-5 with 42-year-old CHW)*

### Effectiveness of the interventions in increasing immunization coverage

In terms of absolute numbers, the number of children that received various antigen increased in both the intervention facilities and control facilities except for MR antigen that declined after the intervention in the intervention facilities and BCG antigen in the control facilities (Table 3).

Table 4 shows the effect of the intervention of coverage of various antigens between the intervention and control using the difference in difference regression model. According to the results from difference in difference analysis, compared to the control facilities, the

**Table 3. Immunization coverage before and after the study implementation between the intervention and control arm.**

| Antigens | Intervention Arm, N = 360 | | Control Arm, N = 155 | |
|---|---|---|---|---|
| | Baseline | Endline | Baseline | Endline |
| DPT3 | 334 (92.8) | 345 (95.8) | 114 (73.5) | 137 (88.4) |
| MR | 378 (105.0) | 320 (88.9) | 106 (68.4) | 116 (74.8) |
| BCG | 270 (75.0) | 291 (80.8) | 120 (77.4) | 117 (75.5) |
| OPV3 | 326 (90.6) | 340 (94.4) | 114 (73.5) | 137 (88.4) |
| PCV3 | 334 (92.8) | 343 (95.3) | 114 (73.5) | 139 (89.7) |
| Rota2 | 275 (76.4) | 348 (96.7) | 117 (75.5) | 139 (89.7) |

intervention caused a 35.1% increase in coverage of BCG antigen (CI: 9.00–61.19). The intervention facilities had a 17.9% increase in DTP3 coverage compared to the control facilities (CI: 1.69–34.1) while for MR, OPV3 and Rota2 antigens, there was no significant effect of the intervention.

## Discussion

This is the first known implementation science study in Uganda on determining the effectiveness of using CHWs to obtain quality and reliable data that can be used for planning and evidence-based response actions. Our findings show that CHWs can be successfully employed to obtain target population data, track, and link defaulters to points of immunization services. The findings further show that holding regular monthly health facility immunization data audit meetings, tracking, and linking of defaulters to immunization services are effective in improving immunization coverage. In addition, equity issues were addressed as neglected children who tend to live in hard-to-reach areas such as forest reserves were reached through house-to-house registration and defaulter tracking through home visits.

This study found that the collected house-to-house population size was higher than the district estimates by 8.4% representing moderate variance which suggests that district projections based on UBOS census estimates are fairly accurate. Therefore, employing CHWs to conduct house-to-house registration may not be cost-effective on a large scale. However, house-to-house registration for eligible population can be used in hard-to-reach areas with consistent low immunization coverage since such areas tend to have existent health inequities. Given the moderate variance, it is also arguable that using the district projections based on the previous population census in Uganda conducted in 2014 [24] to estimate target population may give

**Table 4. Difference in difference regression model of the effect of the intervention on the immunization coverage by antigen.**

| Variables | Vaccine coverage | | | | | |
|---|---|---|---|---|---|---|
| | DPT3 (95% CI) | MR (95% CI) | BCG (95% CI) | OPV3 (95% CI) | PCV3 (95% CI) | Rota2 (95% CI) |
| Period | | | | | | |
| Baseline | 1 | 1 | 1 | 1 | 1 | 1 |
| Endline | 7.47 (-10.29–25.23) | 14.87 (-27.19–59.95) | -15.99 (-36.21–4.21) | 7.47 (-11.21–26.15) | 8.12 (-10.68–26.93) | 22.85 (-7.27–52.99) |
| Treatment | | | | | | |
| Control | 1 | 1 | 1 | 1 | 1 | 1 |
| Intervention | 17.91 (1.69–34.1) ** | 32.09 (-6.31–70.50) | -20.72 (-39.18–2.27) | 15.15 (-1.89–32.21) | 17.91 (0.74–35.07) ** | 9.56 (-17.94–37.07) |
| Period*treatment | -4.82 (-27.76–18.11) | 4.39 (-49.92–58.71) | 35.09 (9.00–61.19) ** | -2.81 (-26.93–21.29) | -5.52 (-29.79–18.75) | -9.97 (-48.88–28.92) |

Note

** represents $p < 0.05$; Period*treatment is the indicator of the intervention effect

increasingly inaccurate estimates with each additional year. Evidence from studies suggests that annual changes in number of surviving infants may vary against the projected national estimates for the target population [25]. Therefore, comparison from alternative independent sources such as data from house-to-house registration can be useful in obtaining an accurate target population (denominator). In areas of consistent low immunization coverage, obtaining accurate target population can be useful to further understand extent of missed opportunities for vaccination.

In this study, the CHWs were given refresher training in EPI service delivery; data management (including review of EPI performance); vaccine management; advocacy, communication, and mobilization. The training lasted for five days, one month before implementation of the study interventions. The CHWs tasks included attending the data audit meetings and conducting home visits to trace and link defaulting children. They were supervised by the health workers involved in EPI service delivery at their respective health facilities to which they were attached. The tasks of the health workers included organizing the data audit meetings; identifying children defaulting at their immunization schedules; reconciling the defaulters who had returned for immunization; and supervising the CHWs.

Data monitoring and use through regular meetings coupled with follow up of these partially immunized children through home visits was key to improving coverage in our study. Results showed that the review meetings provided an opportunity for health workers to analyze, appreciate and use EPI data to make programmatic improvements. These audit meetings just like evidence from another study presented an excellent platform for sharing best practices, lessons learned, dialoguing and providing feedback on results of EPI indicators with health workers involved in immunization service delivery [20]. Evidence also shows that regular immunization data review meetings have been seen as important means to improving the health worker performance especially when conducted among peers with an aim of problem solving. CHWs can learn well from their peers who are at an equivalent level simply because they are able to share their knowledge, ideas and experiences [20].

This study showed that using CHWs to track and link defaulters to points of immunization services was a feasible intervention. In our study, overall, more than two thirds (68%) of defaulters returned for catch up immunization. Evidence from a study conducted in Kenya showed that households visited by CHWs were 1.7 times more likely to have fully immunized children than those that were not [26]. Another similar study conducted in a hard-to-reach area in Kenya also showed that home visits were seen to improve immunization coverage and reduce on inequities brought about by differences in geographical location and socio-economic status [27]. In addition, it can be argued that home visits present an opportunity for the education of caregivers on the importance of immunization. However, for cases where home visits may be not feasible, extending immunization outreaches in the hard-to-reach areas may reduce inequities and improve on the vaccination coverage.

There were several reported challenges experienced by CHWs while conducting home visits including lack of protective wear, trekking long distances and lack of financial incentives or remuneration, among others. Similar findings have been reported in studies conducted in Ethiopia [28] and Zimbabwe [29].

Results from our study further elicited various reasons for children defaulting from their immunization schedules. These included religious beliefs prohibiting uptake of vaccination services; restricted movements due to the COVID-19 pandemic; migration of families for economic activities such as farming; fear of multiple injections; domestic violence, and ignorance of the caregivers on the benefits of immunizations. Similar reasons for defaulting have been reported from studies conducted in; Uganda [30]; Ethiopia [31] and Kenya [32].

We observed significant differences in the immunization coverage between the health facilities in the intervention and control arms for specific antigens at endline. These observed differences may be attributed to the monthly health facility immunization data audits with the identification and followup of defaulters through home visits. A number of studies support our results and demonstrate that defaulter tracking of children through home visits increases immunization coverage [33, 34]. The 'fifth child' project conducted in Ethiopia further demonstrated that use of CHWs to conduct defaulter tracking increased the immunization coverage [35]. Our findings are consistent with most studies demonstrating that CHWs contributed to the increase of immunization coverage [26, 36]. A study conducted in India showed that CHWs were highly cost-effective where percentage of measles vaccination increased by approximately 10% [15].

Overall, at endline, coverage for all antigens (except MR in the intervention arm and BCG in the control arm) increased in both the intervention and control arms. The increase of coverage in the control arm could be explained by the fact that during the same period of implementing of activities in the intervention health facilities, there were post-covid-19 pandemic immunization revitalization activities initiated by the district health office and geared to poorly performing health facilities in Mayuge district of which health facilities in the control arm were a part of. In addition, the significant decline in BCG coverage (Table 4) observed in the control arm at endline could be explained by the low institutional deliveries of 63% [13] as previously mentioned implying that children born outside the health facilities may miss out on the BCG vaccine given at birth.

CHWs can play an important role in attaining universal health coverage by strengthening health systems to provide people centered care that is equitable and culturally appropriate [14]. In our study, CHWs were trained, given specific tasks, and regularly supervised. We believe this contributed to their good performance. Evidence from a review of studies further demonstrates that CHWs are likely to perform well if given; appropriate training and skilling; clear limited tasks; supportive supervision; and appropriate remuneration/incentives [14].

The cost of implementing the intervention package in the three health facilities over a period of three months was $14,900 including the audit meetings and defaulter tracing that cost $1,230 and $2,560 respectively.

This Implementation study had some limitations that need to be acknowledged. There were limitations in obtaining an accurate target population given the possibility that not all under-one-year children (eligible population) were registered during the house-to-house visits. There were five vaccine hesitant households that refused registration and seven households that were found locked as their inhabitants had temporarily migrated to other areas for agricultural purposes such as planting of maize, Therefore, it can be argued that the inaccessible households could have contributed to a higher registered population of under-one-year children during the house-to-house exercise. Another possible limitation could have been contamination between the control and intervention facilities arising from CHWs from the different arms sharing information on the study proceedings in their respective communities. Lastly, the time (three months) for the study implementation was short to observe the meaningful impact of the interventions on the trends of immunization coverage.

## Conclusions

Although it is feasible to use CHWs to obtain accurate population data, it may not be cost-effective to conduct house-to-house registration at full scale since the data obtained was comparable to the district estimates which are based on national census projections. In neglected, hard-to-reach populations and other areas with low immunization coverage, house-to-house

registration, and home visits by the CHWs may be useful in addressing the immunization inequities.

Facilitating monthly health unit immunization data audit meetings for identifying, tracking, and linking defaulters to immunization services are effective means of increasing immunization coverage and equity and should therefore be considered for integration into immunization programs.

## Supporting information

**S1 Dataset.**
(XLSX)

## Acknowledgments

We wish to extend our appreciation to the institutions and individuals that contributed to the implementation of this research study on using community health workers to improve data quality, coverage, and equity of immunization services in Mayuge district, Uganda, that is the Alliance for Health Policy and Systems Research at World Health Organization Head Quarters, Health Support Initiatives, Makerere University School of Public Health, Ministry of Health-Immunization department, Mayuge District Health Office, In-charges of Health Facilities in Bukabooli sub-county, and the field data collection teams.

We wish to thank the district health team members, health facility staff, CHWs who participated in the interviews. We recognize the support of the community members who participated in the study.

## Author Contributions

**Conceptualization:** Pamela Bakkabulindi, Immaculate Ampeire, Marta Feletto, Simon Muhumuza.

**Data curation:** Pamela Bakkabulindi, Simon Muhumuza.

**Formal analysis:** Pamela Bakkabulindi, Lillian Ayebale, Paul Mubiri, Simon Muhumuza.

**Funding acquisition:** Marta Feletto.

**Investigation:** Pamela Bakkabulindi, Lillian Ayebale, Simon Muhumuza.

**Methodology:** Pamela Bakkabulindi, Simon Muhumuza.

**Project administration:** Pamela Bakkabulindi.

**Resources:** Marta Feletto.

**Supervision:** Pamela Bakkabulindi.

**Validation:** Pamela Bakkabulindi.

**Visualization:** Simon Muhumuza.

**Writing – original draft:** Pamela Bakkabulindi, Immaculate Ampeire, Lillian Ayebale, Marta Feletto, Simon Muhumuza.

**Writing – review & editing:** Pamela Bakkabulindi, Immaculate Ampeire, Lillian Ayebale, Paul Mubiri, Marta Feletto, Simon Muhumuza.

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
