## [Decision Letter · Decision Letter 0]

4 Jul 2023

PONE-D-23-13404Engagement of Community Health Workers to Improve Immunization Coverage Through Addressing Inequities and Enhancing Data Quality and Use is a Feasible and Effective Approach: A Cluster Randomized Trial in UgandaPLOS ONE

Dear Dr. Bakkabulindi,

Thank you for submitting your manuscript to PLOS ONE. After careful consideration, we feel that it has merit but does not fully meet PLOS ONE’s publication criteria as it currently stands. Therefore, we invite you to submit a revised version of the manuscript that addresses the points raised during the review process.

I suggest you to revise the paper with the sufficient use of the statistics. Similarly, please make clarity over the study design, is this true RCT, Mixed method study, quasi experimental study or implementation study.

We look forward to receiving your revised manuscript.

Kind regards,

Kanchan Thapa, MPH, MPhil

Academic Editor

PLOS ONE

Journal Requirements:

   "This study was supported by the Alliance for Health Policy and Systems Research (Alliance). The Alliance is able to conduct its work thanks to the commitment and support from a variety of funders. These include Gavi, the Vaccine Alliance contributing designated funding and support for this project, along with the Alliance's long-term core contributors from national governments and international institutions. For the full list of Alliance donors, please visit: https://ahpsr.who.int/about-us/funders. "  

7. Please upload a copy of Figure 2, to which you refer in your text on page 10. If the figure is no longer to be included as part of the submission please remove all reference to it within the text.

8. Please include your tables as part of your main manuscript and remove the individual files. Please note that supplementary tables (should remain/ be uploaded) as separate "supporting information" files

Additional Editor Comments:

Dear Authors,

I enjoyed reading your paper. However, I feel there are sufficient rooms for improvement in the paper. Please find the attached comments for your kind reference.

Please take care of grammar. It starts from the first line of the Abstract. Please also make clarity about intervention and control group in the abstract.

Formatting/referencing is another issues in the paper. Please take care of the formatting requirement of plos One Journal.

Is contribution to the literature section required in PLOS One Journal? Please take care of it.

The title suggest that the paper is purely an epidemiological study with lots of rigorous statistics, however there are qualitative parts inside. That is also also the beauty of the study. However, I suggest to represent the both qualitative and quantitive aspect in the tile or keep the title in general format. Stating the RCT in the title causes the biases in the study.

Line 68-70. Is EPI launched in all over the world or Uganda?

Generally, mentioning the affiliation of UN agencies in paper requires the consent from their host agencies. Does Marta Feletto have the consent to represent WHO in this paper?

In result section, is there any opportunities to mention SD or SE to represent the differences? If yes, please mention.

I saw little use of statistics to show the Effectiveness of the interventions in increasing immunization coverage. If there is any opportunities to compare the effectiveness through the use of data, please mention.

I found little information about the results from Table 2,3 and 4.

You have frequently talked about equity, but I did not see any difference in terms of data in the paper, please mention all these things.

I am confused with implementation study, RCT or mixed method study?

Also, take care of all the reveiers' comments.

Reviewers' comments:

Reviewer's Responses to Questions

**Comments to the Author**

1. Is the manuscript technically sound, and do the data support the conclusions?

Reviewer #1: Yes

Reviewer #2: No

2. Has the statistical analysis been performed appropriately and rigorously? 

Reviewer #1: Yes

Reviewer #2: No

3. Have the authors made all data underlying the findings in their manuscript fully available?

Reviewer #1: Yes

Reviewer #2: Yes

4. Is the manuscript presented in an intelligible fashion and written in standard English?

Reviewer #1: Yes

Reviewer #2: Yes

5. Review Comments to the Author

Reviewer #1: Peer Review: PONE-D-23-13404

This CRT is a well thought-out and well-written piece with a robust background, clear methodology, analysis and discussions. Very minor comments below.

Line 143 – please add some basic data including demographics of Mayuge district.

Line 164 – please provide more elaborate description on the statement “Health facilities were randomized into intervention and control arm respectively, based on the administrative immunization coverage for MR..”

Reviewer #2: 1. How the author calculate and test chi-square when did data analysis using MS-Excel 2013?

2. Was the difference between House to house registration versus the projected district population for under one-year children among intervention and control arms significant (7.9% vs 7.5%)?

3. How the author measured effectiveness without doing any modelling?

4. The author can do difference-in-difference analysis to compare immunization coverage between intervention and control arms?

6. PLOS authors have the option to publish the peer review history of their article (what does this mean?). If published, this will include your full peer review and any attached files.

Reviewer #1: **Yes: **Ibrahim Dadari

Reviewer #2: **Yes: **Anisuddin Ahmed

---

## [Author Response · Author response to Decision Letter 0]

5 Sep 2023

Response to Reviewer Comments

PONE-D-23-13404

Engagement of Community Health Workers to Improve Immunization Coverage Through Addressing Inequities and Enhancing Data Quality and Use is a Feasible and Effective Approach: An Implementation Study in Uganda

PLOS ONE

Dear Kanchan Thapa

PLOS ONE

Thank you for considering our manuscript for publication in PLOS ONE and for the decision to submit a revised version of the manuscript. This rebuttal letter responds to each point raised by the reviewers.

We would also request for waiver of publication fees considering that the research was carried out in a developing country (Uganda).

We would like to clarify that the funders had no role in the study design, data collection and analysis, decision to publish, or preparation of the manuscript. 

We have uploaded our raw data and analyzed set as a file along with the manuscripts and cover letter. 

We have added a new author Paul Mubiri who assisted in the re-analysis and reporting of data. 

We have made revisions to 6 references ensuring that they are in Vancouver style and with traceable/current links to PDFs.

Academic Editor

Comment 1

Thank you for submitting your manuscript to PLOS ONE. After careful consideration, we feel that it has merit but does not fully meet PLOS ONE’s publication criteria as it currently stands. Therefore, we invite you to submit a revised version of the manuscript that addresses the points raised during the review process.

I suggest you to revise the paper with the sufficient use of the statistics. Similarly, please make clarity over the study design, is this true RCT, Mixed method study, quasi experimental study or implementation study.

Response 1

Thank you for your response and consideration. We have clarified the study design to reflect an implementation study. We have re-done the analysis to reflect difference in difference analysis as advised by the reviewers below. This re-analysis is further expounded on in the responses below.

Reviewer #1

Journal Requirements

Comment 1

Response 1

Thank you for the suggestions. We have reviewed the style requirements and complied accordingly. 

Comment 2 

Please note that funding information should not appear in any section or other areas of your manuscript. We will only publish funding information present in the Funding Statement section of the online submission form. Please remove any funding-related text from the manuscript.

Response 2

Thank you for this suggestion. We have removed the funding information from the main manuscript text.

Comment 3

Thank you for stating the following financial disclosure: 

 "This study was supported by the Alliance for Health Policy and Systems Research (Alliance). The Alliance is able to conduct its work thanks to the commitment and support from a variety of funders. These include Gavi, the Vaccine Alliance contributing designated funding and support for this project, along with the Alliance's long-term core contributors from national governments and international institutions. For the full list of Alliance donors, please visit: https://ahpsr.who.int/about-us/funders . " 

Response 3

Comment 4

In your Data Availability statement, you have not specified where the minimal data set underlying the results described in your manuscript can be found. PLOS defines a study's minimal data set as the underlying data used to reach the conclusions drawn in the manuscript and any additional data required to replicate the reported study findings in their entirety. All PLOS journals require that the minimal data set be made fully available. For more information about our data policy, please see http://journals.plos.org/plosone/s/data-availability.

Response 4

Thank you for the comment. We have uploaded both the raw and analyzed data sets. 

Comment 5

We note that you have stated that you will provide repository information for your data at acceptance. Should your manuscript be accepted for publication, we will hold it until you provide the relevant accession numbers or DOIs necessary to access your data. If you wish to make changes to your Data Availability statement, please describe these changes in your cover letter and we will update your Data Availability statement to reflect the information you provide.

Response 5

Thank you for the comment. We have uploaded both the raw and analyzed data sets. 

Comment 6

Your ethics statement should only appear in the Methods section of your manuscript. If your ethics statement is written in any section besides the Methods, please move it to the Methods section and delete it from any other section. Please ensure that your ethics statement is included in your manuscript, as the ethics statement entered into the online submission form will not be published alongside your manuscript.

Response 6

Thank you for the comment. The Ethics section has been to the methods section on Page 13-14, Line 311-316. 

Comment 7

Please upload a copy of Figure 2, to which you refer in your text on page 10. If the figure is no longer to be included as part of the submission please remove all reference to it within the text.

Response 7

Thank you for the suggestion. The figure referred to in the text has been removed and is not referred to anymore in the text.

Comment 8

Please include your tables as part of your main manuscript and remove the individual files. Please note that supplementary tables (should remain/ be uploaded) as separate "supporting information" files

Response 8

Thank you for this comment. We have inserted the tables within the main manuscript and have accordingly uploaded the supplementary tables as supporting files. 

Comment 9

Response 9

Thank you for the comment. We have made revisions to 6 references by ensuring that the style is Vancouver and those with links to pdfs, we have revised one reference to ensure that it is current and with a traceable link.

Reviewer #2

Minor comments that should be addressed

Comment 1

Dear Authors,

I enjoyed reading your paper. However, I feel there are sufficient rooms for improvement in the paper. Please find the attached comments for your kind reference.

Please take care of grammar. It starts from the first line of the Abstract. Please also make clarity about intervention and control group in the abstract.

Formatting/referencing is another issues in the paper. Please take care of the formatting requirement of plos One Journal.

Is contribution to the literature section required in PLOS One Journal? Please take care of it.

The title suggest that the paper is purely an epidemiological study with lots of rigorous statistics, however there are qualitative parts inside. That is also also the beauty of the study. However, I suggest to represent the both qualitative and quantitive aspect in the tile or keep the title in general format. Stating the RCT in the title causes the biases in the study.

Response 1

1. Thank you for the comments and suggestions. The grammar in the abstract and the entire manuscript has been revised. 

2. Clarity of the intervention and control group in the abstract has been made (lines 36-40, page 2).

3. The references have been revised as mentioned in response 9 above.

4. Contributions to the literature section has been removed.

5. As mentioned above, the title has been revised to reflect that it was an implementation study. It now read as ‘Engagement of Community Health Workers to Improve Immunization Coverage Through Addressing Inequities and Enhancing Data Quality and Use is a Feasible and Effective Approach: An Implementation Study in Uganda’ (Line 1-3, Page 1)

Comment 2

Line 68-70. Is EPI launched in all over the world or Uganda?

Response 2

Thank you for this comment. The launch is global, and the sentence has been revised to read: Since the global launch of the Expanded Program on Immunization (EPI) in 1974, immunization coverage has exceeded 80% among infants in several countries (line 65-67, Page 4). 

Comment 3

Generally, mentioning the affiliation of UN agencies in paper requires the consent from their host agencies. Does Marta Feletto have the consent to represent WHO in this paper?

Response 3

Thank you for this observation. Marta Feletto did not receive any objection to be part of the authors.

Comment 4

In result section, is there any opportunities to mention SD or SE to represent the differences? If yes, please mention.

Response 4

Thank you for the comment. Analysis has been re-done and logistic regression to test difference in difference analysis done and reported on as seen below:

Table 4 shows the effect of the intervention of coverage of various antigens between the intervention and control using the difference in difference regression model. According to the results from difference in difference analysis, compared to facilities that did not receive the intervention, intervention caused a 35.1% increase in coverage of BCG antigen (CI: 9.00 – 61.19). The intervention facilities had a 17.9% increase in DTP3 coverage compared to the control facilities (CI: 1.69 – 34.1) while for MR, OPV3, and Rota2 antigens, there was no significant effect of the intervention (Line 472-478, Page 21).

Comment 5

I saw little use of statistics to show the Effectiveness of the interventions in increasing immunization coverage. If there is any opportunities to compare the effectiveness through the use of data, please mention.

Response 5

Thank you for the comment. Analysis has been re-done and logistic regression to test difference in difference analysis done and reported on as seen below:

Table 4 shows the effect of the intervention of coverage of various antigens between the intervention and control using the difference in difference regression model. According to the results from difference in difference analysis, compared to facilities that did not receive the intervention, intervention caused a 35.1% increase in coverage of BCG antigen (CI: 9.00 – 61.19). The intervention facilities had a 17.9% increase in DTP3 coverage compared to the control facilities (CI: 1.69 – 34.1) while for MR, OPV3, and Rota2 antigens, there was no significant effect of the intervention (Line 472-478, Page 21).

Comment 6

I found little information about the results from Table 2,3 and 4

Response 6:

Thank you for the comments. Re-analysis has been done and the tables have since changed. There are currently four tables that are all mentioned in the text.

Refer to the supplemental information section:

Table Legends

S1 This is the S1 Table title. House-to-house registration versus the projected district population for children under-one-year in Bukabooli sub-county, Mayuge district, Uganda

S2 This is the S2 Table title. Number of defaulter children identified, followed up, and returned for immunization services in the Intervention health facilities

S3 This is the S3 Table title. Immunization coverage before and after the study implementation between the intervention and control arm

S4 This is the S4 Table title. Difference in Difference regression model of the effect of the intervention on the immunization coverage by antigen

Comment 7

You have frequently talked about equity, but I did not see any difference in terms of data in the paper, please mention all these things

Response 7:

Thank you for the comment. Equity in this manuscript means reaching children who tend to be missed out by reason of their location. The following are equity statements within the manuscript:

‘The CHWs reported that home visits advanced equity as they were able to reach the hard-to-reach households including vulnerable population’ (line 461-462, page 20).

‘In addition, equity issues were addressed as neglected children who tend to live in hard-to-reach areas such as forest reserves were reached through house-to-house registration and defaulter tracking through home visits’ (line 490-492, Page 22). 

‘However, for cases where home visits may be not feasible, extending immunization outreaches in the hard-to-reach areas may reduce inequities and improve on the vaccination coverage’ (line 535-537, page 24).

Comment 8:

I am confused with implementation study, RCT or mixed method study?

Response 8:

Thank you for the comment. This has been rectified in the title and under study design to reflect an implementation study where health facilities were randomized to two groups: intervention and control as seen in the section below:

‘Study design 

The study was an implementation study with a two-arm cluster randomization design of health facilities in Bukabooli sub-county, Mayuge district. Three health facilities were randomized to receive the intervention package while two received no intervention’ (line 157-160, page 7).

Comment 9:

Also, take care of all the reviewers' comments.

Response 9

Thank you for the comments. All the reviewers' comments have been responded to. 

Review Comments to the Author

Comment 1

Reviewer #1: Peer Review: PONE-D-23-13404

This CRT is a well thought-out and well-written piece with a robust background, clear methodology, analysis and discussions. Very minor comments below.

Response 1

Thank you for the comments.

Comment 2

Line 143 – please add some basic data including demographics of Mayuge district.

Response 2

Thank you for the comment. Demographic data about Mayuge district has been added as seen below:

‘Study setting

The study was conducted in Mayuge district, located in eastern Uganda with a population of over twenty thousand infants (Fig 1). The district has several hard-to-reach areas including 7 islands which are habitable and a huge forest reserve with people residing there. As of 2020, the district had 12 sub-counties including two councils, (Malongo, Jagusi, Bukatuube, Busakira, Imanyiro, Mpungwe, Baitagombwe, Wairasa, Kityererea, Kigandalo, Buwaya, Bukabooli, Mayuge town council, and Magama town council), 74 parishes and 512 villages. There were 3 health-sub districts and a total of 52 functional health centers (HC) including 1 hospital; 2 health center (HC) IV; 6 HCIIIs and 43 HCIIs’ (line 137-145, Pages 6-7). 

Comment 3

Line 164 – please provide more elaborate description on the statement “Health facilities were randomized into intervention and control arm respectively, based on the administrative immunization coverage for MR..”

Response 3

Thank you for the comment. The description has been made clear to read: 

‘Before randomization, health facilities were stratified into two groups: those with MR coverage of less than 70% and those with MR coverage of above 70% (Fig 2). The stratification ensured a mix of high and low coverage health facilities before randomization and a good balance of the health facility characteristics in each arm’ (line 167-170 page 8).

Comment 4

Reviewer #2: 1. How the author calculate and test chi-square when did data analysis using MS-Excel 2013?

Response 4

Thank you for the comment. This section has been revised to read: 

‘Primary analysis was conducted with the intention to treat analyses at the health facility level. A difference in difference regression model was fitted to the data using STATA version 17. The proportion of children that received the antigen under study was computed against the facility target population by month’ (line 284-287, page 12). 

Comment 5

Was the difference between House to house registration versus the projected district population for under one-year children among intervention and control arms significant (7.9% vs 7.5%)?

Response 5 

Thank you for the comment. There were some slight errors in the variance. The intervention arm variance was 8.6% and that of the intervention arm was 8.1%. The variance showed moderate variation (between 5 and 10%) in both arms. This section reads as:

‘Overall, 2,048 children were registered by the CHWs through house-to-house registration as compared to 1,889 according to district estimates. This represents an overall 8.4% variation that is moderate in nature. In the intervention facilities, the variance between the house-to-house population and estimated district population was at 8.6% which is comparable to the 8.1% variance seen in the control facilities’ (line 320-324, page 14). 

Comment 6

How the author measured effectiveness without doing any modelling?

Response 6

Thank you for the comment. Analysis has been re-done and logistic regression to test difference in difference analysis done and reported on as seen below:

Table 4 shows the effect of the intervention of coverage of various antigens between the intervention and control using the difference in difference regression model. According to the results from difference in difference analysis, compared to facilities that did not receive the intervention, intervention caused a 35.1% increase in coverage of BCG antigen (CI: 9.00 – 61.19). The intervention facilities had a 17.9% increase in DTP3 coverage compared to the control facilities (CI: 1.69 – 34.1) while for MR, OPV3, and Rota2 antigens, there was no significant effect of the intervention (Line 472-478, Page 21).

Comment 7

The author can do difference-in-difference analysis to compare immunization coverage between intervention and control arms?

Response 7

Thank you for the comment. Analysis has been re-done and logistic regression to test difference in difference analysis done and reported on as seen below:

Table 4 shows the effect of the intervention of coverage of various antigens between the intervention and control using the difference in difference regression model. According to the results from difference in difference analysis, compared to facilities that did not receive the intervention, intervention caused a 35.1% increase in coverage of BCG antigen (CI: 9.00 – 61.19). The intervention facilities had a 17.9% increase in DTP3 coverage compared to the control facilities (CI: 1.69 – 34.1) while for MR, OPV3, and Rota2 antigens, there was no significant effect of the intervention (Line 472-478, Page 21).

Reviewer's Responses to Questions

Comments to the Author

Comment 1

Is the manuscript technically sound, and do the data support the conclusions?

Reviewer #1: Yes

Reviewer #2: No

Response 1

Thank you for your comments. Analysis has been re-done. 

Comment 2

Has the statistical analysis been performed appropriately and rigorously?

Reviewer #1: Yes

Reviewer #2: No

Response 2

Thank you for your comments. Analysis has been re-done. 

Comment 3

Have the authors made all data underlying the findings in their manuscript fully available?

Reviewer #1: Yes

Reviewer #2: Yes

Response 3

Thank you for your comments. We have made available the raw and analyzed data sets.

Comment 4

Is the manuscript presented in an intelligible fashion and written in standard English?

Reviewer #1: Yes

Reviewer #2: Yes

Response 4

Thank you for the comments. We have corrected the grammatical errors.

---

## [Editor Report · Decision Letter 1]

13 Sep 2023

Engagement of Community Health Workers to Improve Immunization Coverage Through Addressing Inequities and Enhancing Data Quality and Use is a Feasible and Effective Approach: An Implementation Study in Uganda

PONE-D-23-13404R1

Dear Dr. Bakkabulindi,

We’re pleased to inform you that your manuscript has been judged scientifically suitable for publication and will be formally accepted for publication once it meets all outstanding technical requirements.

Kind regards,

Kanchan Thapa, MPH, MPhil

Academic Editor

PLOS ONE

Additional Editor Comments (optional):

Dear Authors,

Thanks for improving the paper as per the reviewers and editor comments. I would like to request you to have some minor correction over fonts, formatting, referencing, addition or deletion of Text Box for IDI and add more information for IDI in a way that [IDI-3, CHW, 45 years, Kampala] etc.

Furthermore, table 3 please make the consistent use of N or n. Please verify and use as per the need. Similarly, remove NO. column in Table 3 and make the consistent table throughout the paper. Your all table require to follow the standard style as per PLOS One criteria. Table 4 has unnecessary lines in between rows. Please correct and delate that.
---

## [Editor Report · Acceptance letter]

9 Oct 2023

PONE-D-23-13404R1 

Engagement of Community Health Workers to Improve Immunization Coverage Through Addressing Inequities and Enhancing Data Quality and Use is a Feasible and Effective Approach: An Implementation Study in Uganda 

Dear Dr. Bakkabulindi:

I'm pleased to inform you that your manuscript has been deemed suitable for publication in PLOS ONE. Congratulations! Your manuscript is now with our production department. 

Kind regards, 

on behalf of

Mr. Kanchan Thapa 

Academic Editor

PLOS ONE